# Effect of Austempering below and above Ms on the Microstructure and Wear Performance of a Low-Carbon Bainitic Steel

**Zhirui Wei, Haijiang Hu \* , Man Liu, Junyu Tian and Guang Xu \***

The State Key Laboratory of Refractories and Metallurgy, Wuhan University of Science and Technology, Wuhan 430081, China; weizhirui@wust.edu.cn (Z.W.); liuman@wust.edu.cn (M.L.); tianjunyu@wust.edu.cn (J.T.)
\* Correspondence: huhaijiang@wust.edu.cn (H.H.); xuguang@wust.edu.cn (G.X.)

**Abstract:** The microstructure and wear performance of a low-carbon steel treated by austempering below and above martensite start temperature (Ms) were investigated. The results show that the bainite, fresh martensite (FM) and retained austenite (RA) were observed in samples austempered above Ms. Except for the three above phases, the athermal martensite (AM) was also observed in samples austempered below Ms. The bainite transformation was accelerated and finer bainite was obtained due to the AM formation in samples austempered below Ms. In addition, the strength and hardness were improved with the decrease of the isothermal temperature and time, whereas the total elongation decreased with the increasing isothermal time and the decreasing isothermal temperature. Moreover, the materials austempered below Ms exhibited better wear performance than the ones treated above Ms, which is attributed to the improved impact toughness by the finer bainite and the enhanced hardness by AM. The best wear resistance was obtained in the samples austempered at 300 °C below Ms for 200 s, due to the highest hardness and considerable impact toughness.

**Keywords:** isothermal time; bainite; athermal martensite; mechanical property; wear performance

## 1. Introduction

Low-temperature bainite steels with ultrafine bainite and film-like retained austenite (RA) have been widely applied in many fields [1–5]. So far, many studies have been conducted on the isothermal transformation at a temperature as low as possible, yet above martensite start temperature (Ms), to obtain finer bainitic microstructure and good mechanical properties [6–9]. It is well known that lower transformation temperature can lead to finer bainite laths. Apart from the transformation temperature, the effect of isothermal time on the microstructure and mechanical properties of low-carbon bainitic steels above Ms was also widely investigated [10–12]. The results indicate that the ultimate tensile strength increases with the increasing isothermal time, while the elongation obviously decreases. The thickness of bainite tends to be increased with the increasing austempering time.

Regarding the bainitic transformation austempered below Ms, some investigations have been conducted [13,14]. Novarro et al. [15] declared that the microstructure consisted of the lath-like bainite, athermal martensite (AM), fresh martensite (FM) and retained austenite (RA) at a isothermal transformation below Ms, and pointed out that AM presented the lath morphology with the characteristic of a sharp tip, whereas the FM showed laths with wavy boundaries and ledge-like protrusions. In addition, Samanta et al. [16] investigated the kinetics of the isothermal transformation of a low-carbon steel below Ms and claimed that the Zener–Hillert model was too slow to explain the observed kinetics at such a low temperature. Recently, Tian et al. [17] compared the effect of austempering temperature below and above Ms on the microstructure and mechanical properties of a bainitic steel, and manifested that although the AM formation below Ms obviously accelerated the transformation kinetics and produced finer bainite, the product of tensile strength and elongation was not improved.

Although many works on the bainite transformation below and above Ms have been performed, the effect of isothermal time on the bainite transformation below Ms is rare. In addition, the wear resistance of bainite steels is especially good at low temperature slightly above or below Ms, and the effect of the isothermal temperature above and below Ms on the wear performance is unclear. Therefore, it is meaningful to investigate the effect of the isothermal time on the microstructure and wear performance of a bainitic steel treated by austempering below and above Ms. The results provide the reference for the design of treating the technology of low-carbon bainitic wear resistance steels.

## 2. Materials and Methods

The chemical composition of the material is Fe-0.20C-1.80Si-1.90Mn-1.00Cr-0.25Mo-0.03V (wt.%). The addition of manganese improves the stability of austenite and prevents the decomposition of austenite during the cooling process [18]. Chromium was added to promote the formation of lower bainite [19]. Molybdenum improves the hardenability of test steel [20,21]. The appropriate silicon was added to hinder carbide precipitation [22]. The tested steels were cast into 50-kg ingots using a vacuum furnace and then hot rolled to a 12 mm plate. The cylindrical samples of Φ10 mm × 86 mm were machined from the hot rolled plate. Thermal simulation tests were conducted on a Gleeble-3500 simulator (DSI lnc., St. Paul, MN, USA). In order to accurately determine the isothermal temperature and quantify the volume fraction of martensite, one sample was directly quenched to room temperature to measure the Ms and the total net dilation of martensite at room temperature. The Ms was measured as 320 °C based on the dilation curve during the quenching process (Figure 1). The heat treatment procedures are given in Figure 2. The specimens were heated at 5 °C/s to 900 °C and austenitized for 300 s to obtain full austenite. After austenization, a part of samples was fast-cooled to 340 °C, which was above Ms (320 °C), held for 200 s and 400 s, and specimens were termed as A200 and A400, respectively. Other samples were fast-cooled to 300 °C, held for 200 s and 400 s, and the specimens were termed as B200 and B400, respectively. Finally, all the isothermally treated samples were fast-cooled to ambient temperature at 20 °C/s.

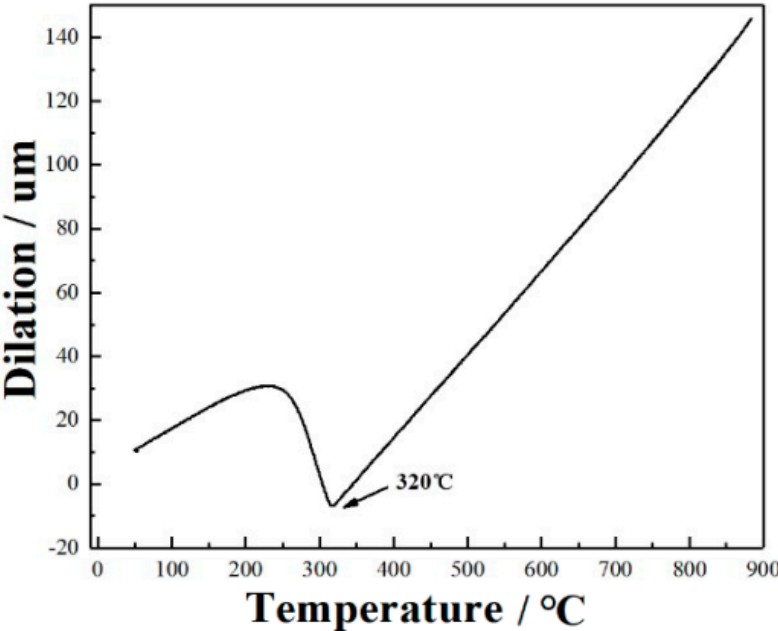

**Figure 1.** Temperature verses dilation during cooling process.

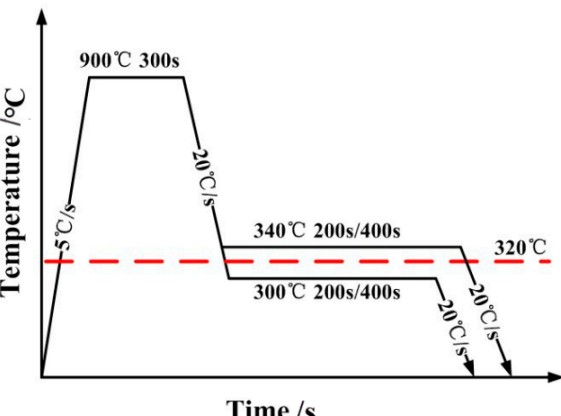

**Figure 2.** The experimental procedure.

The volume fraction of RA ($V_{RA}$) in different samples was calculated according to the integrated intensities of (200) and (211) peaks of ferrite and (200), (311) and (220) peaks of austenite, based on Equation (1) [23]:

$$V_i = \frac{1}{1 + G(I\alpha/I\gamma)} \tag{1}$$

where $V_i$ is the volume fraction of the austenite for each peaks, $I_\alpha$ and $I_\gamma$ are the integrated intensities of the ferrite and austenite peaks. The *G* values for each peak in [17] were used.

Various characterization techniques were used to evaluate the microstructure and properties of different samples. The microstructure of samples after the heat treatment and the morphology of samples after the wearing test were observed using a ZEISS optical microscopy (OM, ZEISS, Oberkochen Germany) and a Nova400 Nano field-emission scanning electron microscopy (SEM, HITACHI, Tokyo, Japan). Tensile tests were conducted on a UTM-4503 electronic universal machine (RIJING LTD, Ningbo, China) at ambient temperature with the cross speed of 1 mm/min. The impact tests were conducted using an impact test machine with V-notch samples. The tensile and impact tests of each sample were repeated three times. The volume fractions of RA were determined using an X-ray diffractometer (XRD, RIGAKU, Japan) under Co-Kα radiation at 40 kV and 150 mA, and the step size was 0.06 deg. The wear tests were carried out on a MLD-10 impact abrasive wear machine (BAOHANG LTD, Zhangjiakou, China) and repeated three times for every sample. The schematic diagram of the impact test machine and specific test parameters are displayed in Figure 3. The set of impact energy was 4 J.

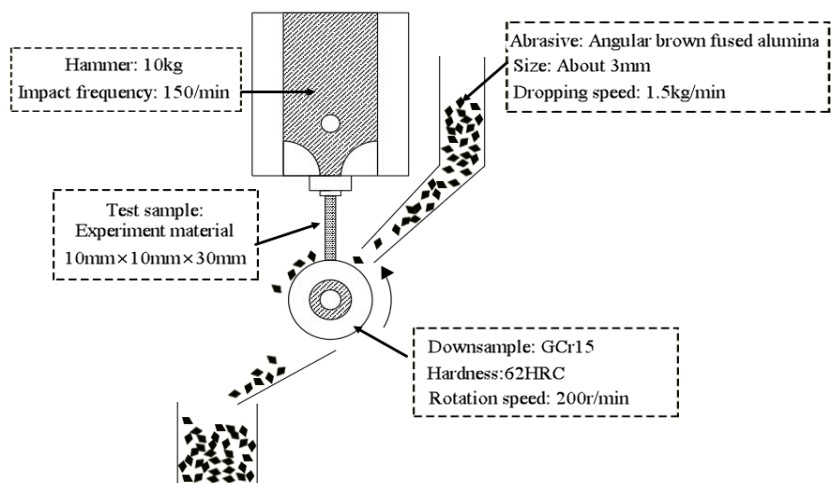

**Figure 3.** Schematic diagram of the impact-abrasive wear machine.

## 3. Results and Discussion

### 3.1. Dilation

Figure 4a presents the dilation–temperature curves at the cooling process of samples austempered above Ms. The undercooled austenite firstly transforms to bainite during the isothermal process, leading to the vertical increase in dilation curves. Then the residual austenite decomposes to FM and M/A islands during the cooling process. Figure 4b displays the dilation–temperature curves of samples austempered below Ms. The AM transformation firstly occurs during the cooling from Ms to the isothermal temperature, and the following vertical curves are corresponded to the bainite transformation during the isothermal holding. According to the results in [24–26], the isothermal martensite cannot be formed in steel with such low carbon content; therefore, it can be concluded that only bainite is formed during the isothermal holding. After the isothermal treatment, the residual austenite transforms to the FM at different extents, and some stable austenite is retained as RA.

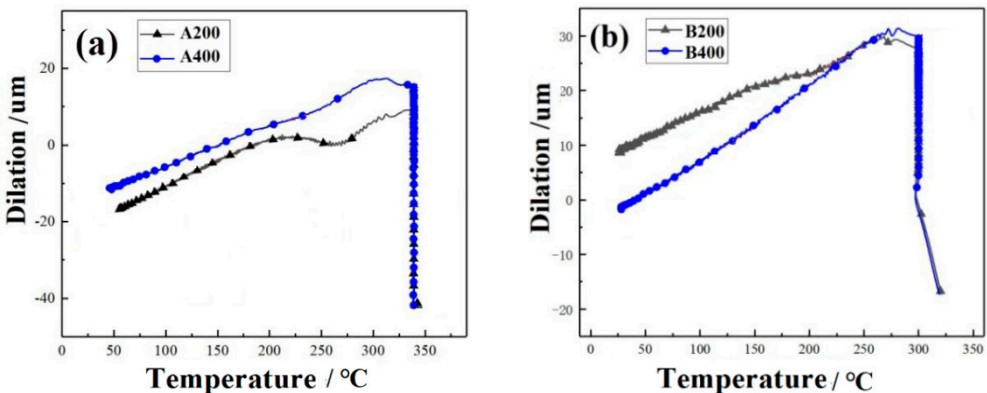

**Figure 4.** Dilation–temperature curves of cooling process: (**a**) isothermal temperature above Ms; and (**b**) isothermal temperature below Ms.

It can be observed from Figure 4 that the dilation caused by the bainite transformation at a longer isothermal time is larger than that at a shorter isothermal time, while the transformation of the FM is greater in the samples at a shorter isothermal time. The more obvious inflexion caused by FM transformation in the dilation curves of shorter isothermal time is mainly attributed to two main reasons: the amount of the residual austenite for the fresh martensite transformation is much more at a shorter isothermal holding time, and the residual austenite is unstable because of the lower carbon emission from bainite. In addition, it can be observed in the curves that the Ms of FM decreases with the extension of the isothermal time because of the stable residual austenite.

The dilation–time curves of the cooling process are displayed in Figure 5. Figure 5a presents the dilation versus time curves of the samples austempered below Ms. The steep slope of the samples B200 and B400 at the first stage is caused by AM formation due to displacive transformation [27]. Then, the slope of the curves changes to flat, meaning that the transformation changes to a bainite transformation. Figure 5b displays the dilation caused only by the bainite transformation during the isothermal holding process. The amount of bainite formation is less in the samples austempered below Ms. This is because part of the undercooled austenite transforms to AM at isothermal temperature below Ms. In addition, the bainite formation rate greatly increases in the samples B200 and B400, as shown in Figure 5c. The AM formation provides more heterogeneous nucleation sites for the subsequent bainite transformation. Moreover, the interior stress caused by AM formation increases the driving force for bainite formation. Therefore, it is reasonable that the bainite transformation rate increases at the isothermal temperature below Ms.

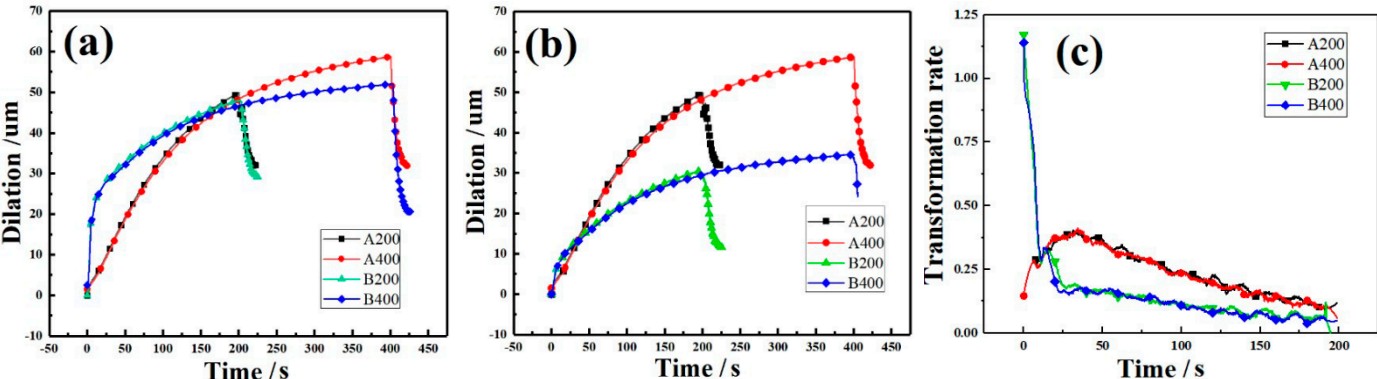

**Figure 5.** Dilation–time curves: (**a**) the total dilation of bainite and martensite; (**b**) the dilation of isothermal transformation; and (**c**) bainite transformation rate.

*3.2. Microstructure*

Figure 6 presents the microstructure of different samples austempered below and above Ms. Figure 6a,b displays typical SEM microstructures of samples below Ms, which consists of the lath-like bainite, AM, FM and RA, while Figure 6c,d show the bainite, FM and RA in samples above Ms. The FM includes martensite/austenite (M/A) and large blocky martensite with a smooth surface [28]. The film-like RA (fRA) is formed alongside the bainite. The morphology of AM is lath with the sharp-tip characteristic [16]. The morphology of FM is convex, while AM is concave, and this is because the FM is more difficult to be etched than the AM, due to the supersaturated carbon. The mean lineal intercept ($\overline{L}$) was measured in the direction perpendicular to the trace of the habit plane of the bainitic ferrite plates from randomly selected locations of SEM micrographs, in order to estimate the true thickness (t) using the following relation [29]:

$$t = 2\overline{L}/\pi \tag{2}$$

The bainite thickness of B200, B400, A200 and A400 was measured to be $21 \pm 4$ nm, $25 \pm 5$ nm, $33 \pm 3$ nm and $36 \pm 5$ nm, respectively. It is obvious that the laths of bainite are finer in the samples at the lower transformation temperature under the same isothermal time. The formation of AM and the lower transformation temperature enhance the strength of the undercooled austenite, leading to the finer bainite laths [30,31].

The volume fractions of RA were calculated to be 4.5% and 2.6% for the specimens B200 and B400, and 5.1% and 3.5% for the specimens A200 and A400, respectively. To quantify the volume fraction of the AM, a method proposed by Tian [32] was used. Figure 7a displays the total net dilation of martensite in a direct quenching sample. In addition, the volume of retained austenite in the direct quenching sample is approximately 0.3%, which can be negligible. Hence, the total net dilation is caused by martensite in the direct quenching sample. Figure 7b presents the total net dilation of martensite at 300 °C, and the volume of martensite in the total dilation at a different austempering temperature can be calculated using the lever rule. The volume fraction of AM at 300 °C is calculated as 32.4%. In order to quantify the volume fraction of FM, the method proposed by Navorro-Lopez [33] was used. The net dilatation at room temperature (Figure 7a) of all samples is compared with the total dilation. Net dilations of FM in B200 and B400 are shown in Figure 7c and signed by D and E, respectively. The volume fractions of bainite can be calculated by balancing the volume fractions of AM, FM and RA in the final cooling to room temperature. The volume fractions of all phases determined for all samples are shown in Figure 7d.

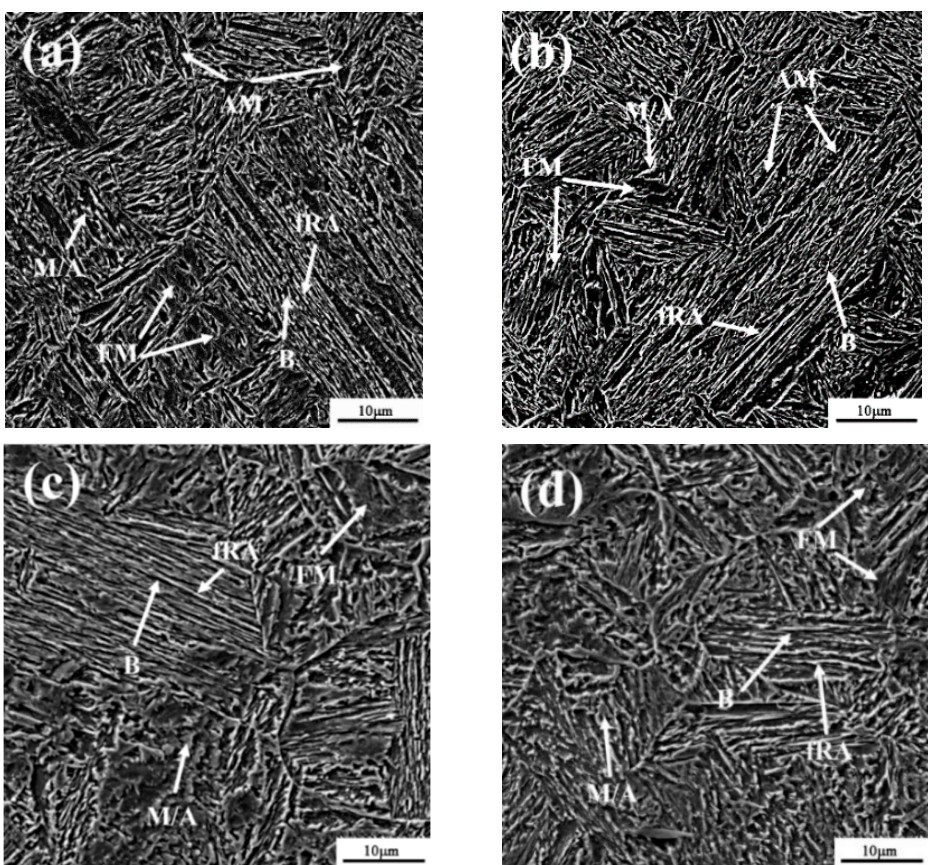

**Figure 6.** SEM micrograph of different samples: (**a**) B200, (**b**) B400, (**c**) A200 and (**d**) A400.

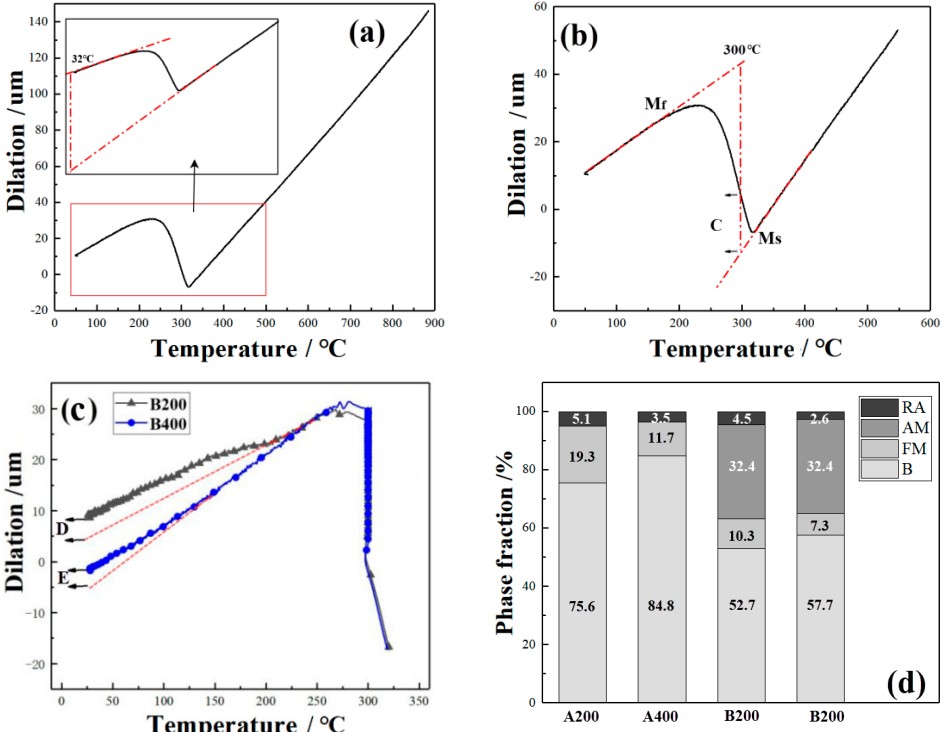

**Figure 7.** (**a**) The dilation curves of direct quenching sample; (**b**) the example of lever rule; (**c**) the example of net dilation of B200 and B400; and (**d**) the volume fraction of phases.

The results show that the volume fraction of the bainite greatly increases with the extension of isothermal time as well as with the increase of the isothermal holding temperature. The formation of the AM consumes more austenite at the lower transformation temperature; thus, less residual austenite is retained for the following bainite formation. In addition, the amount of FM decreases with the extension of isothermal time. This is because more undercooled austenite transforms to bainite at longer isothermal time and the residual austenite becomes more stable due to the carbon emission. The volume fraction of RA decreases with the extension of the isothermal holding time. With the extension of isothermal time, most undercooled austenite transforms to bainite, resulting in only a little retained austenite. On the other hand, at a shorter isothermal time, more undercooled austenite after isothermal holding results in more FM and RA in the final microstructure.

### 3.3. Mechanical Properties

The engineering stress–strain curves for different samples are presented in Figure 8 and the mechanical properties are given in Table 1. All samples exhibit a higher ultimate tensile strength (UTS) and yield strength (YS) because of the fine microstructure of martensite and bainite compared to the steel with a similar composition [17], and the fine microstructure of martensite and bainite is not only attributed to the lower isothermal temperature but also the formation of AM. The refinement of bainite laths with decreasing temperature has been reported in many studies when isothermal above Ms [30,34], but the refinement is further improved in samples austempered below Ms due to the formation of AM before bainite transformation. This fact leads to the initial formation of finer bainite laths due to the fragmentation of initial austenite and the strengthening of untransformed austenite as a consequence of the dislocations induced by AM [35], which further increase the UTS and YS of samples austempered below Ms. Hence, the UTS and hardness of the sample below Ms are higher compared to the sample austempered above Ms.

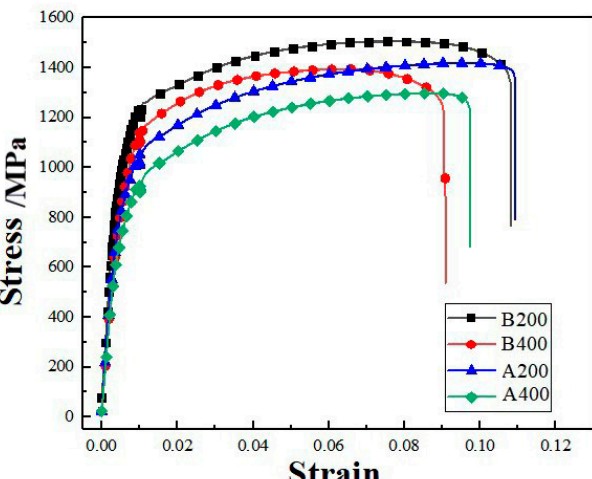

**Figure 8.** Stress–strain curves of different samples.

**Table 1.** Mechanical properties of different samples subjected to different isothermal holding times.

|  | UTS/MPa | YS/MPa | TE/% | Hardness/HV | Impact Toughness/J |
|---|---|---|---|---|---|
| A200 | 1404 ± 10 | 1084 ± 9 | 11.1 ± 0.61 | 455 ± 5 | 38.0 ± 2.0 |
| A400 | 1307 ± 15 | 925 ± 13 | 9.7 ± 0.42 | 441 ± 7 | 39.5 ± 3.5 |
| B200 | 1506 ± 17 | 1269 ± 15 | 10.8 ± 0.56 | 472 ± 4 | 60.5 ± 5.5 |
| B400 | 1421 ± 9 | 1185 ± 10 | 9.1 ± 0.32 | 458 ± 8 | 64.5 ± 4.0 |

The YS and hardness of the samples at a longer isothermal time are smaller than those of the samples at a shorter isothermal time for samples austempered above Ms, which is primarily attributed to less FM in the samples A400. However, for samples austempered

below Ms, the difference in FM is not so large. The tempering of AM strongly affects the yielding behavior [36,37]. Compared to B200, more carbon atoms diffused from the initial carbon-supersaturated AM into the surrounding untransformed austenite during the isothermal process [38], which entails the softening of samples and, as a consequence, the decrease of the overall YS of the longer isothermal B400 sample.

For samples austempered below Ms, the total elongation (TE) of sample B200 is slightly increased compared to B400. The error determining the austenite fraction by XRD is typically high, in the order of 1%, especially in steels with a low austenite fraction. Hence, the effect of RA on TE cannot be accurately analyzed. The TE decreased in the samples austempered below Ms at a longer isothermal time, because the tempering of AM plays an important role. The carbon concentration in the solid solution within AM is reduced during tempering due to carbide precipitation and carbon partitioning into other phases [35]. More carbides at a longer isothermal time might be the reason for the decrease of TE. In addition, the similar amount of RA leads to the similar effect of RA on the mechanical properties. In addition, the morphologies and stability of RA affect the mechanical prosperities, which should be considered in future study.

The −40 °C impact toughness is quite similar at different isothermal times of samples austempered above Ms. The increased and softer bainite increases the impact toughness, but the increased harder FM decreases the impact toughness. These two phenomena lead to similar impact toughness. For samples austempered below Ms, the increase of the impact toughness at longer isothermal time is attributed to not only the increase of bainite amount but also the tempering of AM. The tempering process decreases the residual stress of the AM and then increases the impact toughness [39]. On the other hand, the −40 °C impact toughness is obviously smaller at higher temperature, and the higher isothermal temperature results in coarse bainite laths, which is harmful to the impact toughness. In addition, more FM at a higher temperature also decreases the impact toughness.

The wear properties are mainly related to the hardness and fracture toughness of the samples, and the fracture toughness is proportional to the impact toughness [40]. Hence, higher hardness and impact toughness of samples below Ms result in better wear properties; thus, it can be inferred that the best wear properties are obtained in samples austempered below Ms. Moreover, the comprehensive mechanical propensities of B200 is obviously better than B400, which indicates that the best wear performance might be obtained in samples B200.

### 3.4. Wear Performance

The wear process can be considered as a three-body wear process, and the material removes in multiple ways: wear debris, cracks, flakes and microcutting. Different wear morphologies reflects different wear properties. The abrasive particles may extrude into or cut on the worn surface during the impact process, which results in wear debris and microcutting [41]. Hence, the wear debris and microcutting are more likely to form on low hardness samples. Cracks and flakes are easily located in high stress locations, or other subsurface defects during wearing. The lower impact toughness samples are more likely to be impacted out of defects, and the defects would become the origins of the cracks. Similarly, the cracks in the low impact toughness are easy to propagate along the wear direction and then form flakes [42]. Figure 9 presents the worn surface morphology of different samples. The wear debris in the samples A200 and A400 are attributed to the lower hardness. Moreover, the decrease of impact toughness by thicker bainite leads to more cracks in the A200 and A400 compared to B200 and B400. Hence, the wear performance is worse in the samples austempered above Ms. Liu [40] reported similar results in bainitic steels, that higher isothermal temperature increases the amount of RA but decreases the stability of austenite. A higher amount of RA decreases the hardness, and the martensite induced by unstable austenite is more likely to generate cracks and then decrease the impact toughness, which both decrease the wear performance. More FM in the A200 increases the hardness; thus, the wear debris in the A200 is less than the A400. Regarding the samples

austempered below Ms, the smaller flakes in the B200 are observed because of the finer bainite. The microcutting also decreases in the B200 due to more FM compared to the B400. The wear performance is better in samples austempered at a shorter isothermal holding time. The best wear performance is obtained in the sample B200.

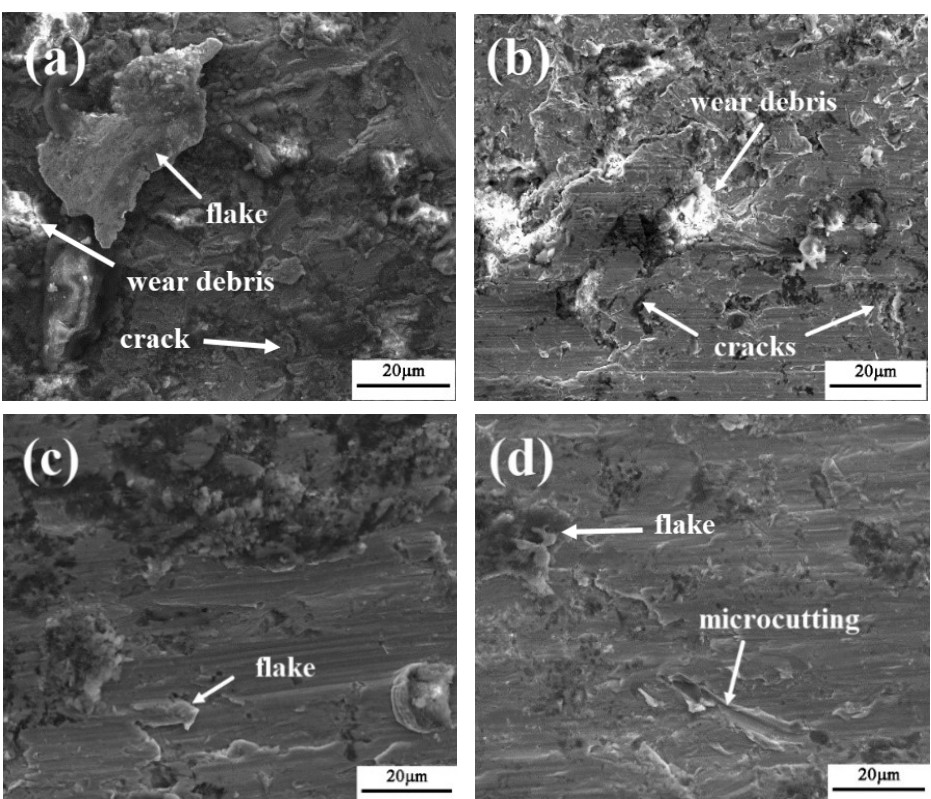

**Figure 9.** Worn surface of different samples: (**a**) A200; (**b**) A400; (**c**) B200; and (**d**) B400.

The curves of mass loss during six wear cycles and total mass loss verses wear time are presented in Figure 10. The mass loss increases with the isothermal holding time and temperature. The better wear resistance is related to the enhanced hardness and the improved impact toughness. For samples austempered at the higher temperature (A200 and A400), the thicker bainite obviously decreases the impact toughness, resulting in a increase of mass loss. With the extension of the isothermal holding time, the amount of bainite increases, whereas the martensite and retained austenite both decrease. The impact toughness is slightly improved at a longer transformation time, but the hardness decreases. Hence, the mass loss increases at a longer isothermal time. The sample treated for 200 s below Ms (B200) exhibits the best wear performance. The mass loss is consistent with wear morphology.

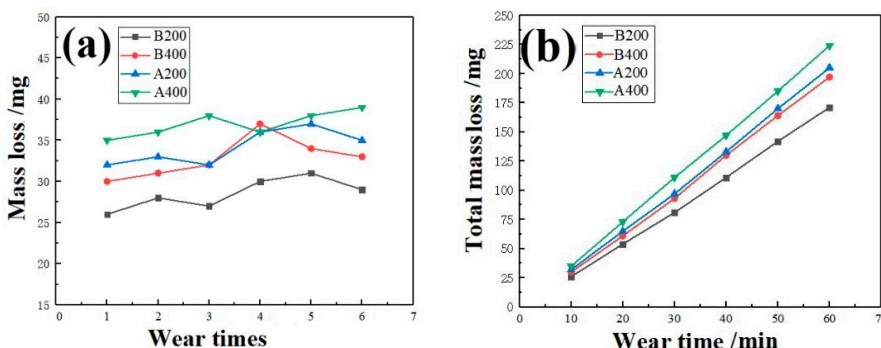

**Figure 10.** (**a**) Mass loss during six wear cycles; and (**b**) total mass loss verses wear time curves.

## 4. Conclusions

The effects of the isothermal time on the bainite transformation, microstructure and wear performance of a low-carbon bainite steel were investigated by different isothermal times at austempering temperatures above and below Ms. The main conclusions are summarized as follows:

1. The YS and hardness were improved with the decrease of isothermal time, whereas the TE slightly decreased owing to the decrease of the bainite. The YS, hardness and impact toughness all decreased at a higher isothermal temperature.
2. The wear performance was better in samples austempered below Ms than above Ms, due to the increase of impact toughness by finer bainite and the improvement of hardness by AM. The mass loss increased with the increase of isothermal time due to the decrease of hardness. Hence, the best wear performance was obtained in the sample isothermally treated at 300 °C for 200 s (B200), which had the highest hardness and considerable impact toughness.
3. The bainite, FM and RA were observed in samples austempered above Ms. Except for the above three phases, the AM was observed in samples austempered below Ms. Moreover, the bainite transformation was accelerated by the formation of AM.

**Author Contributions:** Z.W., conducted experiments, analyzed the data and wrote the paper; H.H., designed experiments and analyzed the data; M.L., conducted experiments and analyzed the data; J.T., analyzed the data; G.X. supervisor, conceived and designed the experiments. All authors have read and agreed to the published version of the manuscript.

**Funding:** The authors gratefully acknowledge the financial supports from the National Nature Science Foundation of China, grant number [51874216, 52104381], the National Nature Science Foundation of Hubei, grant number [2021CFB127], the China Postdoctoral Science Foundation, grant number [2021M702539], the Major Project of Science and Technology of Guangxi (AA 19254009) and the Key Project of Science and Technology of Liuzhou Municipal Government (2019AC10602).

**Institutional Review Board Statement:** Not applicable.

**Informed Consent Statement:** Not applicable.

**Data Availability Statement:** The data presented in this study are available on request from the corresponding author. The data are not publicly avaliable due to an ongoing study.

**Conflicts of Interest:** The authors declare no conflict of interest.

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
