# Peer review of "Effect of Austempering below and above Ms on the Microstructure and Wear Performance of a Low-Carbon Bainitic Steel"

_metals, doi:10.3390/met12010104_

Round 1

Reviewer 1 Report

The authors provide an interesting experimental study on the wear behavior of bainitic steels subjected to isothermal treatments above and below email. There is a decent number of publications discussing the effect of isothermal treatments above and below Ms on the microstructure-mechanical properties relationship. In particular, the effect of isothermal holding time on microstructure and tensile properties has been already studied in these steels. It is fair to acknowledge that fewer studies have devoted investigations on the wear resistance of these steels and this is the main novelty of the paper that has to be better exploited. The discussion in the paper of these microstructure/properties relations is very weak or inexistent as the authors provide results of the performed tests and barely discuss them. The authors should provide a minimum discussion including the results of other works. The authors missed relevant publications in the bibliography regarding this topic.

The microstructural changes in terms of the size of bainite are not included, which in the introduction is stated to play an important role.

Besides, the following minor comments should be addressed:

  • Please revise the numbering of the references in the text and the spelling of the author’s names. Several mistakes are detected.
  • Explain how austenite fractions are obtained in the experimental section, not in the results.

Author Response

Response to Reviewer 1 Comments

Dear reviewer,

We appreciate your professional comments on our manuscript. The comments are very helpful for us to improve the quality of our work. We have made all necessary revisions according to reviewer’s comments, and the detailed modifications are explained as follows point by point.

Point 1:The discussion in the paper of these microstructure/properties relations is very weak or inexistent as the authors provide results of the performed tests and barely discuss them. The authors should provide a minimum discussion including the results of other works. The authors missed relevant publications in the bibliography regarding this topic.

Response: Thank you for your professional comment. The discussion on the microstructure/properties relations and the results of other works are added in the revised manuscript in line 208-217, page 7, line 221-225, page 8, highlighted by red color.

Point 2:The microstructural changes in terms of the size of bainite are not included, which in the introduction is stated to play an important role.

Response: The microstructural changes in terms of the size of bainite is described in line 150-153, page 5, highlighted by red color. The effect of bainite microstructural changes on wear performance is described in line 219-220, page 8, line 227-228, page 8, highlighted by red color.

Point 3: Please revise the numbering of the references in the text and the spelling of the author’s names. Several mistakes are detected.

Response: The numbering of the references in the text and the spelling of the author’s names are checked and revised.

Point 4: Explain how austenite fractions are obtained in the experimental section, not in the results.

Response:

The explanation on how austenite fractions are obtained is moved to the experimental section according to your suggestion in line 80-85, page 3, highlighted by red color.

Thank you very much again for your time and professional comments.

Best regards!

Sincerely yours,

Guang Xu

Reviewer 2 Report

The paper deals with an important topic and well fits with the Metals topics.

Below some suggestions:

  1. Line 25: add the following recent references (as reference n.4 and n.5): a) Di Schino A., Analysis of phase transformation in high strength low alloyed steels. Metallurgija, 2017, 56, 349-352. b) Di Schino A., Gaggiotti M., Testani C., Heat treatment effect on microstructure evolution in a 7% Cr steel for forging. Metals, 10, 2020, 808.
  2. Line 190: I do not understand what authors mean: "this is example 1 of an equation": please rewrite
  3. Line 220: any explanation for that?

Author Response

Response to Reviewer 2 Comments

Dear reviewer,

We appreciate your professional comments on our manuscript. The comments are very helpful for us to improve the quality of our work. We have made all necessary revisions according to reviewer’s comments, and the detailed modifications are explained as follows point by point.

Point 1:Line 25: add the following recent references (as reference n.4 and n.5): a) Di Schino A., Analysis of phase transformation in high strength low alloyed steels. Metallurgija, 2017, 56, 349-352. b) Di Schino A., Gaggiotti M., Testani C., Heat treatment effect on microstructure evolution in a 7% Cr steel for forging. Metals, 10, 2020, 808.

Response: The relative references are added according to your suggestion in line 25, page 1, line 275-277, page 9, highlighted by red color.

Point 2:Line 190: I do not understand what authors mean: "this is example 1 of an equation": please rewrite

Response: Sorry for our carelessness. The relative description is rewritten in line 190, page 7, highlighted by red color.

Point 3:Line 220: any explanation for that?

Response: The relative explanation is given in line 208-217, page 7, line 221-225, page 8, highlighted by red color.

Thank you very much again for your time and professional comments.

Best regards!

Sincerely yours,

Guang Xu

Round 2

Reviewer 1 Report

Thanks for addressing some of the points in the previous revision. Unfortunately, I still consider the discussion on mechanical properties is still very weak. I have added more points where the discussion can be improved and extensive references that can be added to support the discussion.

It is obvious that the laths of bainite are finer in the samples at the lower transformation temperature --> it is not that obvious, provide some estimation on the size or mark in the micrographs what do you consider bainite lath width. See this work on determining lath size in bainite by SEM:

  • Garcia-Mateo, J.A. Jimenez, B. Lopez-Ezquerra, R. Rementeria, L. Morales-Rivas, M. Kuntz, et al., Analyzing the scale of the bainitic ferrite plates by XRD, SEM and TEM. Mater Charact. 122 (2016) 83–89.

The formation of AM and lower transformation temperature enhance the strength of the undercooled austenite, leading to the finer bainite laths --> add a reference here.

The discussion of the mechanical properties should be further improved

The engineering stress-strain curves for different samples are presented in Figure 8 and the mechanical properties are given in Table 1.

All samples exhibit high ultimate tensile strength (UTS) and yield strength (YS) because of the fine microstructure of martensite and bainite --> compared to what? Are these typical values of bainitic steels? Please, provide a comparison with other works on the topic. What is the effect of finer microstructures in some of the treatments as claimed by the authors? See e.g.

- J. Tian, G. Xu, M. Zhou, H. Hu, Refined bainite microstructure and mechanical properties of a high-strength low-carbon bainitic steel treated by austempering below and above Ms, Steel Research Inter (2018) 1–10.

The YS and hardness of the samples at shorter isothermal time are higher than those of the samples at longer isothermal time, which is primarily attributed to the more FM in the samples A200 and B200 à in the case of above Ms treatments is a very plausible reason, but below Ms the difference in FM is not that big. The authors have not considered and discussed the effect of AM tempering explaining these decreases in yield strength and hardness in below Ms specimens. See e.g.

  • Navarro-López, J. Hidalgo, J. Sietsma, M.J. Santofimia, Unravelling the mechanical behaviour of advanced multiphase steels isothermally obtained below M,    Des.  188  (2020)  108484

Moreover, the increase of the total elongation (TE) at shorter isothermal holding time is attributed to the more amount of RA --> The error determining austenite fraction by XRD is typically high in the order of 1 %, especially in steels with a low austenite fraction as it is the case. This fact should be highlighted and an in-depth discussion should be initiated on the effect of RA and other microstructural features. 1) Provide references supporting the possible effect of the measured RA fraction variation in the higher UE elongation observed. 2) It is not only RA fraction what counts, but RA stability, nothing is commented on this, is the RA stability assumed the same in both low and high isothermal times? 3) What can be the effect of AM tempering on the UE in the below Ms specimens?

Compared to the sample austempered above Ms, the YS and hardness of the sample below Ms are higher, whereas the TE changes to be lower. More AM increases the YS and the hardness of the samples austempered below Ms, --> Comparing A400 and B400 hardness differences lay within the error, the difference in hardness is only obvious between A200 and B200. Hence, the authors' statement only holds for short isothermal times and emphasizes the effect of AM tempering on the properties of the complex microstructures. This has to be further discussed.

 and less amount of bainite and RA in the samples austempered below Ms decreases the TE --> differences in total elongation are not that obvious comparing below and above Ms counterparts.

The -40 °C impact toughness increases with the isothermal holding time due to more bainite. On the other hand, the higher isothermal temperature results in coarse bainite laths, which can destroy the impact toughness  --> Provide references on this. Mention that decreasing the temperature below Ms would lead in austenite to martensite transformation, thus the likely explanation is bainite. Discuss also the effect of FM on toughness.

All these microstructure mechanical properties relations should be better linked with wear properties. Besides, these references should be included and discussed in the introduction or in the results:

- J.C. Hell, M. Dehmas, S. Allain, J.M. Prado, A. Hazote, J.P. Chateau, Microstructure-properties relationships in carbide-free bainitic steels, ISIJ Int. 51 (2011) 1724–1732.

- X. Tan, Y. Xu, X. Yang, D. Wu, Microstructure-properties relationship in a one-step quenched and partitioned steel, Mater. Sci. Eng. A 589 (2014) 101–111.

- G. Mandal, S.K. Ghosh, S. Bera, S. Mukherjee, Effect of partial and full austenitisation on microstructure and mechanical properties of quenching and partitioning steel, Mater. Sci. Eng. A 676 (2016) 56–64.

- S. Yan, X. Liu,W.J. Liu, T. Liang, B. Zhang, L. Liu, Y. Zhao, Comparative study on microstructure and mechanical properties of a C-Mn-Si steel treated by quenching and partitioning (Q&P) processes after a full and intercritical austenitization, Mater. Sci. Eng. A 684 (2017) 261–269.

- J. Feng, T. Frankenbach, M. Wettlaufer, Strengthening 42CrMo4 steel by isothermal transformation below martensite start temperature, Mater. Sci. Eng. A 683 (2017) 110–115.

- A. Zinsaz-Borujerdi, A. Zarei-Hanzaki, H.R. Abedi, M. Karam-Abian, H. Ding, D. Han, N. Kheradmand, Room temperature mechanical properties and microstructure of a low alloyed TRIP-assisted steel subjected to one-step and two-step quenching and partitioning process, Mater. Sci. Eng. A 725 (2018) 341–349.

- A. Navarro-Lopez, J. Hidalgo, J. Sietsma, M.J. Santofimia, Influence of the prior athermal martensite on the mechanical response of advanced bainitic steel, Mater. Sci. Eng. A 735 (2018) 343–353.

Round 3

Reviewer 1 Report

In this last review, the discussion of the relation microstructure-mechanical properties has been improved and supported by relevant references in the topic. The characterization of bainite features is also been improved by adding the lath dimensions. Hence, all my comments have been covered satisfactorily and thus I consider the work is worth publication in the present form. I would like to thank the authors for this work, adding new insight into the behavior of bainitic steels isothermally treated below Ms.